

# Digital dissection of the pelvis and hindlimb of the red-legged running frog, *Phlyctimantis maculatus*, using Diffusible Iodine Contrast Enhanced computed microtomography (DICE µCT)

Amber J. Collings[1,2] and Christopher T. Richards[2]

[1] School of Science Engineering and Design, Teesside University, Middlesbrough, United Kingdom
[2] Structure and Motion Laboratory, Royal Veterinary College, London, United Kingdom

## ABSTRACT

**Background**. The current study applies both traditional and Diffusible Iodine Contrast Enhanced computed microtomography (DICE µCT) techniques to reveal the musculoskeletal anatomy of *Phlyctimantis maculatus*. DICE µCT has emerged as a powerful tool to visualise intricate musculoskeletal anatomy. By generating 3D digital models, anatomical analyses can be conducted non-destructively, preserving the *in situ* 3D topography of the system, therefore eliminating some of the drawbacks associated with traditional methods. We aim to describe the musculature of the spine, pelvis, and hindlimb, compare the musculoskeletal anatomy and pelvic morphology of *P. maculatus* with functionally diverse frogs, and produce 3D digital anatomy reference data.

**Method**. An adult frog was stained using an aqueous Lugol's solution and scanned in a SkyScan1176 *in vivo* µCT scanner. Scan images were reconstructed, resampled, and digitally segmented to produce a 3D model. A further adult female frog was dissected traditionally for visualisation of tendinous insertions.

**Results**. Our work revealed three main findings: (1) *P. maculatus* has similar gross muscular anatomy to *Rana catesbeiana* (bullfrog) but is distinct from those species that exhibit ancestral traits (leopelmids) and those that are highly specialised (pipids), (2) *P. maculatus*'s pelvic anatomy best fits the description of Emerson's walking/hopping pelvic morphotype IIA, and (3) a split in the semimembranosus and gracilis major muscles is consistent with the reported myology in other anuran species.

**Discussion**. While DICE µCT methods were instrumental in characterising the 3D anatomy, traditional dissection was still required to visualise important structures such as the knee aponeurosis, tendinous insertions, and fasciae. Nonetheless, the anatomical data presented here marks the first detailed digital description of an arboreal and terrestrial frog. Further, our digital model presents *P. maculatus* as a good frog model system and as such has formed a crucial platform for further functional analysis within the anuran pelvis and hindlimb.

Corresponding author
Amber J. Collings,
A.Collings@tees.ac.uk

## INTRODUCTION

Anurans are key to understanding the intricate connections among vertebrate musculoskeletal elements enabling limb motion (*Lombard & Abbott, 1907*; *Kargo & Rome, 2002*; *Kargo, Nelson & Rome, 2002*). As such, frogs have been used as models for understanding the biomechanics of jumping (e.g., *Calow & Alexander, 1973*; *Kargo, Nelson & Rome, 2002*; *Roberts & Marsh, 2003*; *Astley & Roberts, 2014*; *Porro et al., 2017*), swimming (*Gillis & Biewener, 2000*; *Nauwelaerts & Aerts, 2002*; *Richards & Biewener, 2007*; *Clemente & Richards, 2013*) and walking (*Ahn, Furrow & Biewener, 2004*; *Reynaga, Astley & Azizi, 2018*). One particular region of interest in anurans is the morphological variation in the sacrum and pelvis, thought to play a large role in the locomotor versatility observed across anuran taxa (*Emerson, 1979*). Three distinct morphotypes were defined by Emerson, Type I, Type IIA, and Type IIB, differing in muscle origin, insertion, and size, shape of sacral diapophysis, and the nature of the ligamentous attachments. Hypothesised to allow for differential rotations about the ilio-sacral joint, the different morphotypes are understood as specialisations for different locomotor behaviours, such as swimming (Type I), walking (Type IIA), and jumping (Type IIB) (*Emerson, 1979*; *Reilly & Jorgensen, 2011*; *Jorgensen & Reilly, 2013*).

In this study we explore the pelvic and hindlimb anatomy of the hyperoliid *Phlyctimantis maculatus* (*Portik & Blackburn, 2016*) (previously known as *Kassina maculata*). While colloquially called the red-legged running frog, *P. maculatus* excels at walking, running, hopping, climbing, and jumping (*Ahn, Furrow & Biewener, 2004*; *Porro et al., 2017*; *Richards, Porro & Collings, 2017*; *Richards, Eberhard & Collings, 2018*). With muscular hindlimbs, this species forages in the savannah, long grass, and bushland terrestrially (*Bwong et al., 2017*) while also escaping into the trees, climbing and jumping arboreally, making use of their well-developed toepads (*Loveridge, 1976*). Given their proclivity to walking, running, and climbing, we predict this species possesses a Type IIA pelvic morphotype. This 'walking morphotype' is described as having a dorsoventrally flattened sacrum with slight expansion of the diapophyses and triangular ilio-sacral ligaments (*Emerson, 1979*).

Additional to using multiple locomotor modes, *P. maculatus* are easy to work with, robust to maintain in laboratory conditions, and can currently be sourced ethically in the USA/EU/UK. We will use both traditional and emerging 3D techniques to study the musculoskeletal anatomy of this 'multifunctional' frog in detail. Using aqueous iodine to increase the radiopacity of the soft tissues, diffusible iodine contrast enhanced computed microtomography (DICE µCT) allows anatomical analyses to be conducted non-destructively (*Metscher, 2009a*; *Metscher, 2009b*; *Herdina et al., 2010*; *Vickerton, Jarvis & Jeffery, 2013*; *Gignac & Kley, 2014*; *Lautenschlager, Bright & Rayfield, 2014*; *Herdina et al., 2015*; *Porro & Richards, 2017*; see *Gignac et al., 2016* for a full review) and has emerged as a powerful functional morphology tool to visualise intricate musculoskeletal anatomy across diverse systems (For example: *Cox & Jeffery, 2011*; *Jeffery et al., 2011*; *Baverstock, Jeffery & Cobb, 2013*; *Düring et al., 2013*; *Cox & Faulkes, 2014*; *Lautenschlager, Bright & Rayfield, 2014*; *Holliday et al., 2013*; *Gignac & Kley, 2014*; *Kleinteich & Gorb, 2015*; *Klinkhamer et*

al., 2017; Bribiesca Contreras & Sellers, 2017, see Gignac et al., 2016 for a comprehensive review).

Our current work builds upon the first, and only, published DICE μCT description of a frog to date, performed on *Xenopus laevis* (*Porro & Richards, 2017*). While *Xenopus* is regularly used as a model species, they are fully aquatic and specialised swimmers. We therefore present the first detailed digital dissection of a 'multifunctional' terrestrial and arboreal species. By combining virtual techniques with traditional dissection we aim to: (1) describe the locomotor and postural musculature of the spine, pelvis, and hindlimb, (2) contextualise and compare the pelvic morphotype and musculoskeletal anatomy of *P. maculatus* with other functionally diverse frogs, and (3) contribute to the growing collection of 3D digital anatomy data for further use in research and education (See Data S1).

## MATERIALS & METHODS

### Musculoskeletal geometry

#### Diffusible iodine contrast enhanced μCT scanning

One adult *P. maculatus* frog (15.7 g body mass), obtained from Amey Zoo (Hempstead, UK), was euthanised by Tricaine methanesulfonate (MS222) overdose (0.02% MS222, 0.04% $NaHCO_3$) followed by removal of the heart (compliant with primary and secondary methods of amphibian euthanasia as per procedures approved by Home Office License 70/8242). The wound in the chest was closed using non-absorbable braided silk suture (6-0) to limit internal exposure to fixative and staining solution therefore avoiding over-staining. The frog (whole and un-skinned) was fixed in 10% neutral buffered formalin (NBF) (HT501128, Sigma Aldrich) for 29 hours, at room temperature, in a darkened environment. Following the fixation process, any remaining fixative was removed by transferring the specimen in to a PBS solution where it was left soaking overnight, at room temperature. To enhance soft tissue contrast for imaging, predominantly of the muscular anatomy, the frog was stained using an aqueous Lugol's solution (L6146, Sigma Aldrich, a.k.a. iodine potassium iodide, $I_2KI$). To avoid over staining, test scans were performed at regular intervals throughout the staining process. After each test scan the specimen was re-introduced to the stain. Depending on the results of the scan, various recommendations were applied to increase stain perfusion, including skinning the specimen (conducted after 3 days), increasing the stain concentration (conducted after 16.5 days), and injecting stain into body (conducted after 31.5 days) (see Table 1). The frog was placed in 70% pure ethanol (02877; Sigma Aldrich, St. Louis, MO, USA) for preservation during transport to and from the scanner. All μCT scans were conducted on the SkyScan1176 *in vivo* μCT scanner (Bruker microCT, Kontich, Belgium) in the Biological Services Unit of the Royal Veterinary College, Camden campus. The specimen was wrapped in cellophane and taped down for each scan to prevent drying out or movement during imaging. For each of the test scans, a section of the mid-thigh and/or pelvis was chosen for imaging since this was a time efficient way to check image quality of both bone and ample soft tissue. After a total of 39 days, the entire specimen was imaged in the final scan (17.64 μm resolution,

**Table 1 Laboratory parameters for staining and scanning.** The staining regime used was continuous therefore cumulative stain duration refers to the number of days the specimen was exposed to the staining solution in total whereas stain duration details the duration of exposure to the stain in that particular test round.

| Sample | Stain concentration | Stain duration | Cumulative stain duration | Scan results | Recommendation |
|---|---|---|---|---|---|
| Whole, Un-skinned | 7.5% | 3 days | 3 days | No effect on musculature | Skin specimen and stain further |
| Whole, Skinned | 7.5% | 13.5 days | 16.5 days | Stain not fully perfused | Stain further with increased concentration |
| Whole, Skinned | 20% | 7.5 days | 24 days | Stain not fully perfused | Stain further |
| Whole, Skinned | 20% | 7.5 days | 31.5 days | Stain not fully perfused in the thigh | Stain further, with some injection of stain into thighs and body |
| Whole, Skinned | 20% | 7.5 days | 39 days | Sufficient perfusion | Final scan to be conducted using the following settings: 17.64 μm resolution, 50 kV, 362 μA, 1 mm Al filter |

50 kV, 362 μA, 1 mm Al filter). The scan images were then reconstructed using NRecon software (V1.6.10.1; Bruker microCT, Kontich, Belgium), showing the subcutaneous soft tissue topography of *P. maculatus* (Figs. 1A–1C).

### Visualisation and segmentation

The reconstructed DICE μCT scan images from the final scan were resampled (1 in 5) and then visualised and digitally segmented in Amira 6.0.1 (FEI, Hillsboro, OR, USA) (Figs. 2A–2D). The multiplanar viewer function tab was used to create a volume model of all structures. Both the magic wand and paintbrush tools of the segmentation editor function tab were used to assign voxel selections as either bone or muscle material. Voxel assignment was made on the basis of greyscale value, those of the lightest colour denoted bone or muscle, whereas black voxels denoted air space. Due to limitations of the technique (see Discussion), author's discretion and anatomical expertise were required for selection and assignment of voxels at the boundary between two materials. Every individual bone and muscle between the 4th vertebra and the distal digits of the hindlimbs were assigned as a separate material. Muscle and bone identifications were performed with the aid of previously published descriptions of other frog species (*Dunlap, 1960*; *Emerson & De Jongh, 1980*; *Přikryl et al., 2009*). Once the material selections for all muscles were complete, the segmented label field data was resampled (data resampled by 50% in the *Z* direction) before being rendered into 3D surface meshes to produce a 3D representation of the musculoskeletal anatomy of the frog lower spine, pelvis, and hindlimb (Fig. 2D). During surface rendering the file underwent constrained smoothing to minimise the visual appearance of the voxels, providing a more even surface and therefore realistic representation of the tissues. Each material surface mesh was exported individually as an STL file. Using the software 3-Matic (Materialise Inc., Leuven, Belgium), the individually exported meshes corresponding to the bones of the spine, pelvis, hindlimb, and foot, as well as the individual muscles of the left side (spanning from the spine to the tarsometatarsal (TMT) joint) were combined to create a 3D model of *P. maculatus* (Fig. 2D). Additionally, viewable as a 3D PDF, the digital model presents all skeletal material and all muscles of the left side, totalling 17 bones and 41 muscles (Data S1, see Article S1 for 3D PDF user guide).

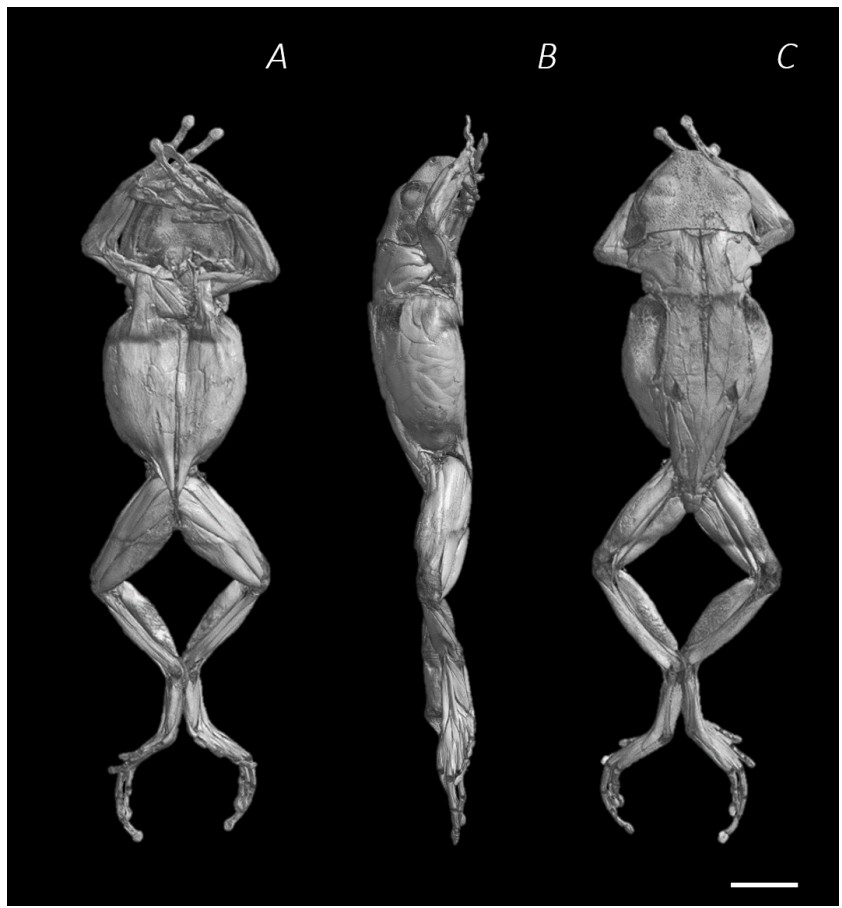

**Figure 1** **Reconstructed DICE μCT scan images of *Phlyctimantis maculatus*.** Created using N-Recon and CT-vox software. (A) Ventral, (B) lateral, and (C) dorsal view. Scale bar in white 1 cm.

The individual metatarsals and phalangeal foot bones are grouped together and referred to as the metatarsals and digits, respectively.

### Traditional dissection

Due to the limitations of DICE μCT to visualise tendinous material (see Discussion), musculoskeletal anatomy data were additionally obtained from a female adult frog by traditional means (specimen *P. maculatus*, body mass: 31.20 g, Source: Amey Zoo, Hempstead, UK). This animal had been previously euthanised using the methods described above, and fixed in 10% NBF (HT501128, Sigma Aldrich) for ~24 hours. Each muscle of the spine, pelvis, and hindlimb (left side only) was identified, described, and removed in its entirety. Identifications were made with the aid of both the digital dissection of the 3D model (current work) as well as the previously published dissection data (*Ecker, 1889*; *Dunlap, 1960*; *Emerson & De Jongh, 1980*; *Duellman & Trueb, 1986*; *Přikryl et al., 2009*). The muscle names used throughout the current study are consistent with previous anuran dissection literature (*Dunlap, 1960*; *Emerson, 1979*; *Emerson, 1982*; *Emerson & De Jongh, 1980*; *Duellman & Trueb, 1986*; *Přikryl et al., 2009*).

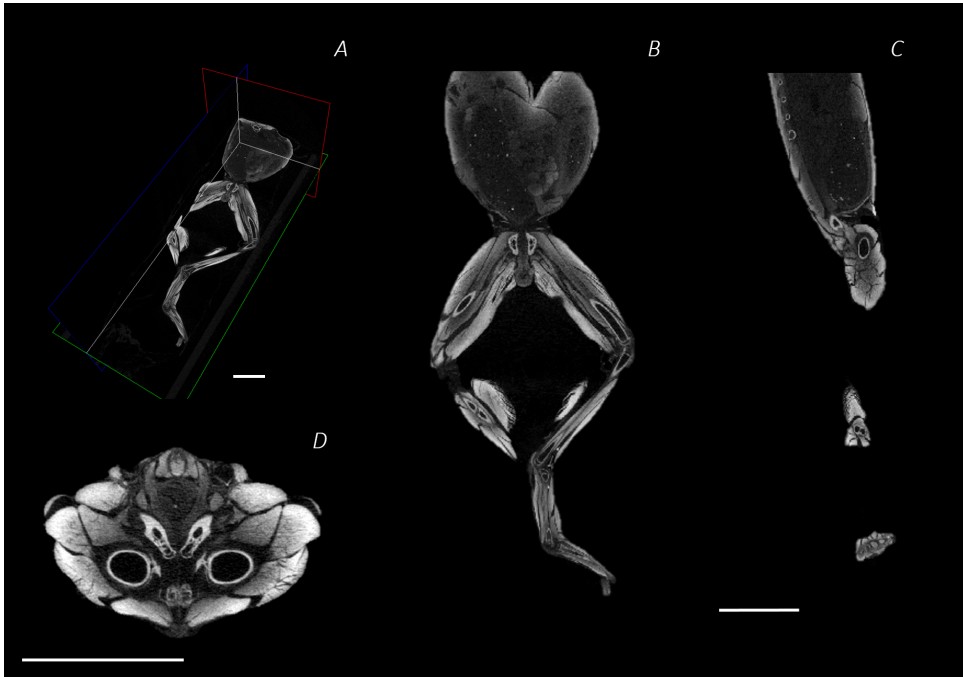

**Figure 2** **Reconstructed DICE μCT scan images showing internal structure of the distal spine, pelvis, and hindlimb of *Phlyctimantis maculatus*.** (A) Posterior-oblique view of slices in the transverse, sagittal, and frontal planes, (B) frontal section through the mid-body, (C) sagittal section through the right side of the body and hindlimb, and (D) transverse section through the body at the hip joint. Scale bars in white, all 1 cm.

## RESULTS

### Musculoskeletal geometry

Using traditional and digital dissection, 41 muscles were identified between the back and the proximal foot. We have grouped muscles as per their anatomical region i.e., the back and pelvis, thigh, shank, tarsals and foot. A summary of the detailed gross anatomy findings, including origins, insertions, and notable features are presented in Table 2. Figures 3–6 present a superficial, medial, deep, and skeletal digital dissection.

Six muscles were identified belonging to the back and pelvic muscles group: the LD, IL, CS, CI, IE, and PY. A fascial sheet, found directly beneath the skin of the dorsal side, covered the entire back, beneath which led the two axial muscles IL and LD, and the two pelvic muscles CI and CS (Figs. 3C; 4C; 7A–7B). The IE muscle led along the lateral surface of the iliac shaft (Figs. 3B– 3C ; 4B; 7B) whereas the PY muscle was observed joining the urostyle tip to the femur (Figs. 3C; 4C).

In total, seventeen muscles made up the muscle mass of the thigh: II, TFL, GL, CR, GR, SM, IFB, IFM, SA, AL, ST, PEC, OE, AM, QF, OI, and GE. Superficially, the dorsal muscle mass of the thigh included the GL and CR (Figs. 3C; 7C). The TFL was also visible and positioned proximal to the CR (Fig. 3B). Together, SM and GR made up the majority of the muscle mass of the ventral thigh (Figs. 3A and 3C; 7C–7D). Both a minor and a major

Collings and Richards (2019), *PeerJ*, DOI 10.7717/peerj.7003

**Table 2 Summary table of gross anatomy of all of the axial, pelvic, and hindlimb muscles analysed from *Phlyctimantis maculatus*.**

| Muscle (Abbreviation) | Origin | Insertion | Notable features |
|---|---|---|---|
| Longissimus dorsi (LD) | Anterior spine and base of skull (atlas and occipital bone) and vertebrae | Along the anterior half of the urostyle | Long muscles, consisting of multiple segments, unified by thin septa, which each originate from individual vertebrae via fleshy connections. |
| Iliolumbaris (IL) | Pre-sacral vertebrae | Medially: sacral diapophysis Laterally: sacroiliac joint and anterior iliac shaft | |
| Coccygeosacralis (CS) | Dorsal sacral diapophysis and proximal urostyle | Urostyle | Roughly triangular in shape and fill the space between the ilia and urostyle. Fleshy attachments. |
| Coccygeoiliacus (CI) | Sacral diapophysis and medial, anterior iliac shaft | Medial surface of urostyle | |
| Pyriformis (PY) | Posterior urostyle | Proximal femur | Present as a small slip of muscle. |
| Iliacus externus (IE) | Lateral surface of iliac shaft | Proximal femur | Narrow and cylindrical muscle with large fleshy origin and tendinous insertion. |
| Iliacus internus (II) | Medial surface of the ilium | Proximal femur | Wraps ventrally around the ilia from origin to insertion. Fleshy attachments. |
| Tensor Fascia Latte (TFL) | Lateral Ilium | Cruralis muscle | Small slip of muscle with soft tissue insertion. |
| Gluteus maximus (GL) | Ilium | Cruralis muscle/Knee aponeurosis | Soft tissue insertion. |
| Cruralis (CR) | Ventral border of the ilium | Knee aponeurosis of anterior surface of the knee joint | Large muscle forming the knee aponeurosis distally. |
| Gracilis major (GR major) | | | Large fleshy muscle separated roughly in half by a connective tissue septum. |
| Gracilis minor (GR minor) | Ischium | Knee aponeurosis medially | Small thin belly that runs along the lateral side of the major belly. |
| Semimembranosus (SM) | Dorsal rim of ischium and ilium | Knee aponeurosis laterally and ventrally | Large fleshy muscle separated roughly in half by a connective tissue septum. |
| Iliofibularis (IFB) | Ilium | Knee aponeurosis laterally | Narrow and cylindrical. |
| Iliofemoralis (IFM) | Ventral border of the ilium | Femur approximately mid-shaft proximo-distally | Narrow and cylindrical. |
| Sartorius (SA) | Ventral border of the ischium | Knee aponeurosis medially | Long strap muscle. |
| Adductor longus (AL) | Ventral border of the ischium | Knee aponeurosis medially | Present as a long strap muscle. |

**Table 2** (*continued*)

| Muscle (Abbreviation) | Origin | Insertion | Notable features |
|---|---|---|---|
| Semitendinosus dorsal head (STd) | Posterior ventral border of the ischium | Tibiofibula ventrally | Two heads with tendinous origins that share a common tendinous insertion. The ventral head passes through the adductor magnus muscle belly. |
| Semitendinosus ventral head (STv) | Posterior dorsal border of the ischium | | |
| Pectineus (PEC) | Ventral border of the ischium | Femur approximately mid-shaft proximo-distally | Twisted muscle belly. Shares fleshy origin with, and inserts slightly proximal to, obturator externus. |
| Obturator externus (OE) | Ventral border of the ischium | Femur approximately mid-shaft proximo-distally | Shares fleshy origin with pectineus. |
| Adductor magnus (AM) | Ventral border of the ischium | Femur distal shaft | Large muscle with two sections, perforated by the ventral head of the semitendinosus. Wraps around the femur almost entirely enveloping the distal third of it. |
| Quadratus femoris (QF) | Ischium | Proximal femur | Interacts closely with gemellus to present as single mass. |
| Obturator internus (OI) | Entire pelvic rim | Proximal femur | Forms a fleshy ring around the hip joint. |
| Gemellus (GE) | Ischium | Proximal femur | Interacts closely with quadratus femoris to present as single mass. |
| Plantaris longus (PL) | Knee aponeurosis posteriorly | Plantar aponeurosis via long tendon | Large, pennate, biarticular muscle with a long tendon that merges with the plantar aponeurosis. |
| Tibialis posticus (TiP) | Posterior surface of tibiofibula | Astralagus | Distally tapered muscle belly with a tendinous insertion. |
| Tibialis anticus longus head 1 (TiAL1) | Knee aponeurosis laterally | Lateral border of the proximal calcaneum | Two distinct heads that are roughly equal in size, sharing a tendinous origin with separate tendinous insertions. |
| Tibialis anticus longus head 2 (TiAL2) | | Medial border of proximal astralagus | |
| Peroneus (PER) | Knee aponeurosis laterally | Distal tibiofibula laterally | Cylindrical muscle covering lateral surface of tibiofibula. |
| Extensor cruris brevis (ECB) | Knee aponeurosis | Anterior medial surface of the tibiofibula | Narrow cylindrical muscle. |
| Tibialis anticus brevis (TiAB) | Anterior surface of tibiofibula | Medial surface of the proximal astralagus | Large fleshy origin covering tibiofibula laterally. |
| Plantaris profundus (PP) | Calcaneal ligament | Plantar aponeurosis | Separate to flexor digitorum brevis superficialis. |
| Tarsalis posticus (TaP) | Calcaneal ligament | Distal astralagus | Roughly rectangular shaped muscle. |
| Flexor digitorum brevis superficialis (FDBS) | Calcaneal ligament | Penetrates into plantar aponeurosis | Thin muscle belly. |
| Transversus plantae proximalis and distalis (TPP and D) | Distal calcaneum and plantar cartilage | Plantar aponeurosis | Unified as one muscle but extremely fragile. |
| Intertarsalis (IN) | Lateral margin of the astralagus and medial margin of the calcaneum | Tendinous insertion at distal union of tarsals | Pennate muscle filling the gap between the elongate tarsal bones. |

Collings and Richards (2019), *PeerJ*, DOI 10.7717/peerj.7003

**Table 2** (*continued*)

| Muscle (Abbreviation) | Origin | Insertion | Notable features |
| --- | --- | --- | --- |
| Extensor digitorum communis longus (EDCL) | Lateral side of distal tibiofibula | Third digit of foot | Long, narrow muscle with tendinous origin in common with tarsalis anticus. |
| Extensor brevis superficialis (EBS) | Dorsal and medial surface of the calcaneum | Tendinous insertions onto the digits of the foot | Multiple bellies with tendinous insertions sharing a common fleshy origin. |
| Adductor brevis dorsalis and plantaris (ABD and P) | Medial surface of calcaneum | Fifth metatarsal and digit | Challenging to separate the two muscle bellies. |
| Tarsalis anticus (TaA) | Lateral side of distal tibiofibula | Dorsal surface of the astralagus | Roughly rectangular shaped with a tendinous origin in common with extensor digitorum communis longus. |
| Adductor prehallucis (AP) | Edge of plantar aponeurosis | Pre-hallux | Small superficial slip of muscle. |
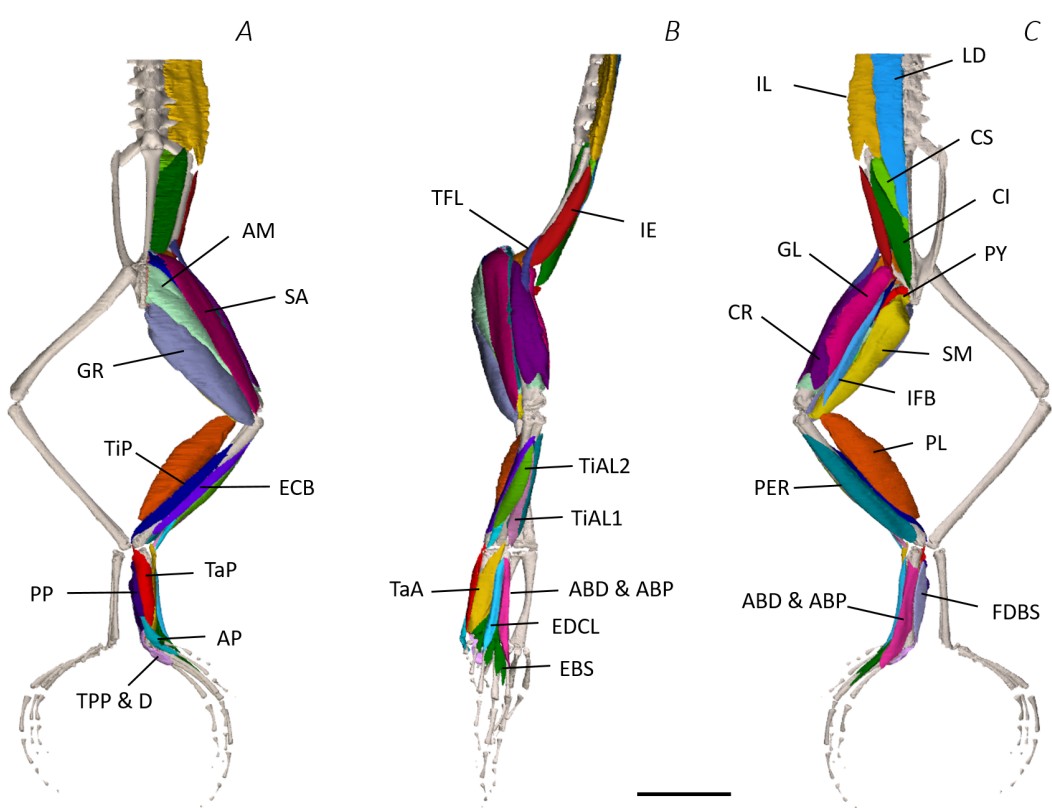

**Figure 3 Superficial digital dissection of the distal spine, pelvis, and hindlimb of *Phlyctimantis maculatus*.** (A) Ventral, (B) lateral, and (C) dorsal view. See Table 2 for muscle abbreviations. For the interactive 3D PDF, see Supplemental Information. Scale bar in black 1 cm.

belly of the GR were observed, the minor appearing only as a thin slip of muscle running along the lateral side of its major counterpart (Fig. 7D). The IFB was positioned between and slightly deep to the GL and SM along the lateral surface of the thigh, while the SA and AM muscles were positioned on the medial surface between the GR and CR (Figs. 3 and 4; 8A–8D). Deeper dissection revealed the II muscle which crossed the hip joint by wrapping around the ventral surface of the ilium (Figs. 4B–4C; 5B–5C). The AL led medial to II and directly beneath SA (Fig. 4B). The ST was split into two heads that ran along the ventral surface of the femur (Figs. 4A and 4C; 8A–8B). The small muscles of the hip joint were deeper still. The IFM muscle was positioned lateral and ventral to II, whereas PEC (a thin muscle with a twisted belly as seen in Fig. 8E) and QF were positioned medial and ventral to II (Fig. 5). Deep to OE, the OI muscle covered the whole lateral portion of the pelvic rim, cupping the hip joint (Fig. 5A). The QF and GE muscles interacted closely with each other, forming a fleshy connection between the posterior rim of the pelvis and the proximal femur (Figs. 5A and 5C).

The six muscles of the shank included the PL, TiP, TiAL, PER, ECB, and TiAB. The PL muscle made up the vast majority of the posterior tibiofibular muscle mass (Figs. 3A and 3C; 8F). Superficially, the PER muscle ran along the lateral border of the tibiofibula (Figs.

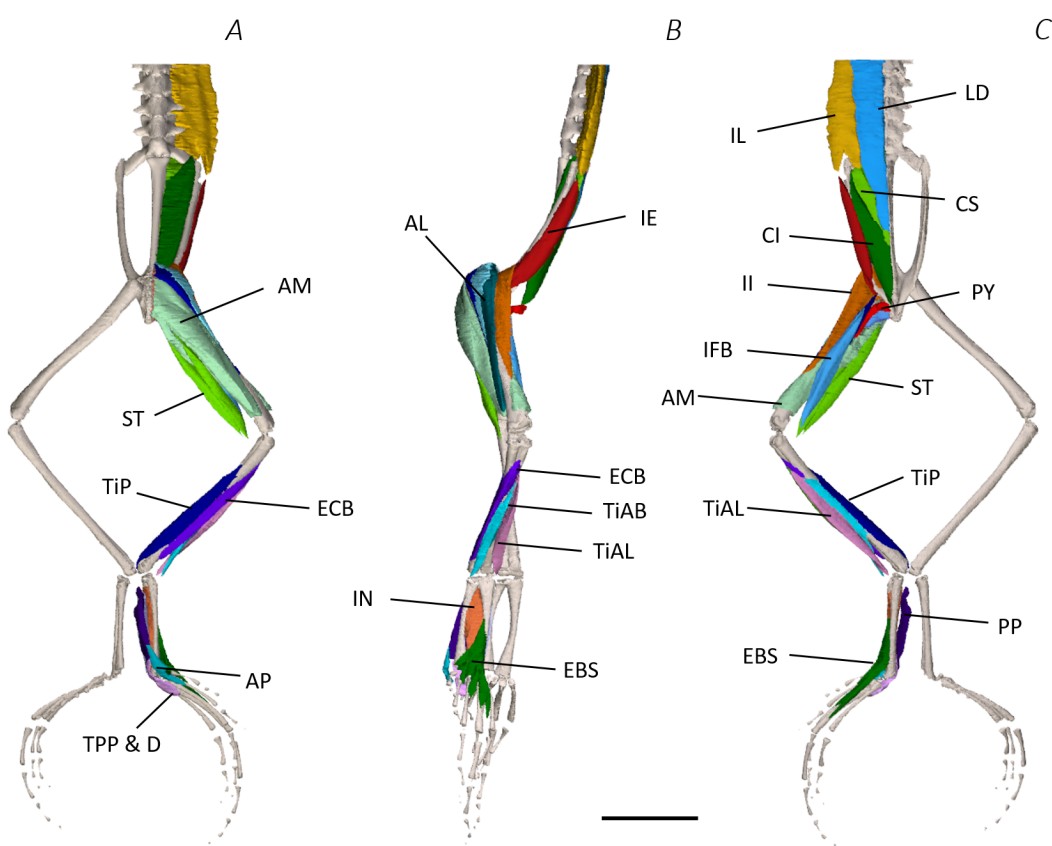

**Figure 4 Medial digital dissection of the distal spine, pelvis, and hindlimb of *Phlyctimantis maculatus*.**
(A) Ventral, (B) lateral, and (C) dorsal view. See Table 2 for muscle abbreviations. For the interactive 3D PDF, see Supplemental Information. Scale bar in black 1 cm.

3C; 7C; 8G), whereas the TiAL muscle, which split into two distinct heads, appeared along the anterior surface of the shank (Figs. 3B; 4B–4C; 8F). Deeper dissection revealed the TiP, ECB, and the TiAB. The TiP was positioned deep to PL and covered the distal two thirds of the posterior surface of the tibiofibula. Medial to TiP was the ECB, wrapping medially from the proximal anterior surface to cover the medial surface of the tibiofibula (Fig. 3A; Figs. 4A–4B; 8F). Finally, TiAB was positioned deep to ECB and the two heads of TiAL (Figs. 4B; 5A–5C).

Twelve muscles were identified belonging to the tarsals and proximal foot, including: PP, TaP, TaA, EDCL, ABD and ABP, FDBS, AP, TPP and D, EBS, and IN. Superficially, the TaA, EDCL, and ABD and P made up the anterior muscle mass of the tarsals, while the posterior muscle mass consisted of the FDBS and PP muscles (Figs. 3A and 3C ; 4C; 5A and 5C). While ABD and ABP were merged in the digital dissection of *P. maculatus* (Figs. 3B–3C) the fragility of these muscles during traditional dissection made it challenging to discern whether or not these muscles were indeed separate or not. The TaP muscle covered the medial portion of the astralagus, superficially (Figs. 3A; 8H), whereas laterally, the calcaneum was covered by ADB and P (Figs. 3B–3C). Distal to the TaP, the AP muscle

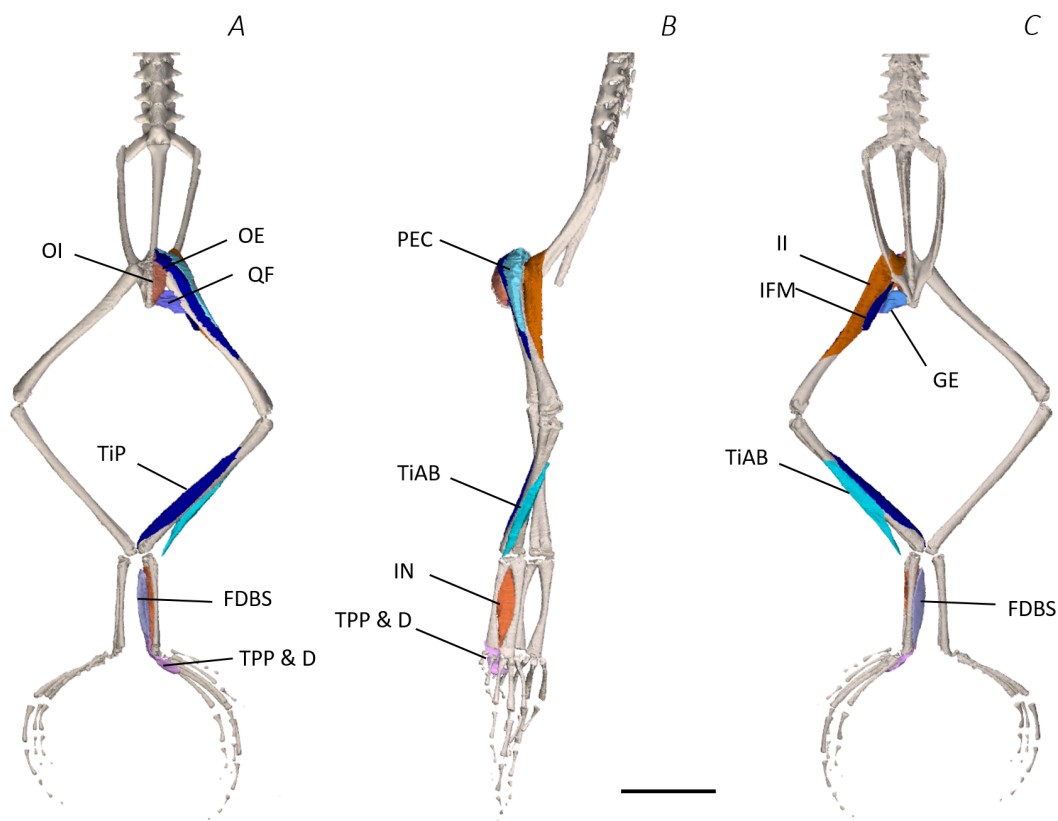

**Figure 5 Deep digital dissection of the distal spine, pelvis, and hindlimb of *Phlyctimantis maculatus*.**
(A) Ventral, (B) lateral, and (C) dorsal view. See Table 2 for muscle abbreviations. For the interactive 3D
PDF, see Supplemental Information. Scale bar in black 1 cm.

crossed the TMT joint (Figs. 3A; 4A; 8H). Deep muscles of this region of the hindlimb
included the EBS, TPP and D, and IN. We have referred to EBS as a single unit however
as can be visualised in Figs. 4B–4C and 8I, while there is a common origin, this muscle
separates distally into multiple separate heads. The TPP and D muscles cover the posterior
surface of the TMT joint (Figs. 3A; 4A; 5A–5B), whereas the IN spans the interosseous gap
between the two elongate tarsal bones (Figs. 4B; 5B).

Notably, connective tissue septa were observed in the two axial muscles, LD and IL, as
well as in two muscles of the thigh, SM and GR. While both the LD and IL consisted of
multiple segments unified by thin septa to form elongate muscle masses (Figs. 9A–9B), in
SM and GR, the connective tissue septum split the muscle bellies approximately in half,
separating the proximal and distal ends (Figs. 9C–9F). In the SM muscle the separation
was a diagonal line from left to right at a slight proximal-distal angle (Figs. 9C and 9D),
whereas in the GR the separation was a 'U' shaped line running through the middle of the
muscle belly from left to right (Figs. 9E and 9F). In the digital scan reconstruction images,
the separations in GR and SM were visible as thin radiolucent darker lines transecting the
middle of the muscle bellies (Figs. 9D and 9F). As in the axial muscles, the separations in
these muscles appeared to interrupt the parallel muscle fibres.

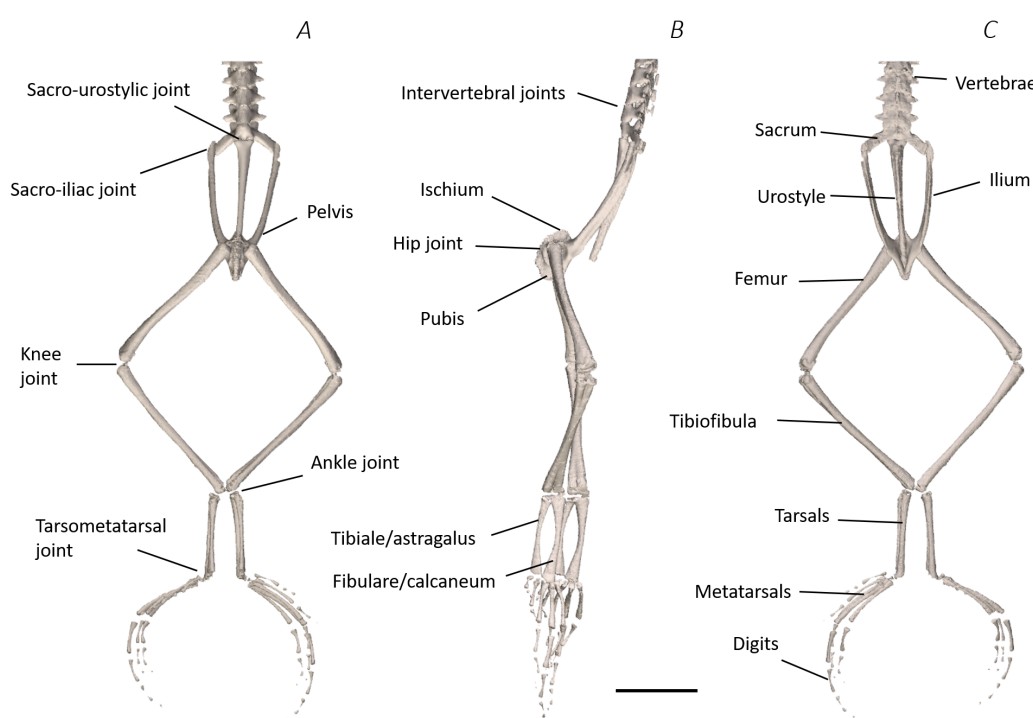

**Figure 6** **Skeletal digital dissection of the distal spine, pelvis, and hindlimb of *Phlyctimantis maculatus.*** (A) Ventral, (B) lateral, and (C) dorsal view. See Supplemental Information for the interactive 3D PDF. Scale bar in black 1 cm.

### Pelvic morphotype verification

Comparison of the traditional and digital dissection findings with descriptions of Emerson's three morphotypes revealed that *P. maculatus*'s pelvis best fits the description of morphotype IIA (Fig. 10). The sacrum was dorsoventrally flattened with some lateral diapophyseal expansion, and ligaments appeared triangular in shape. Additionally, the sacro-iliac joint of *P. maculatus* allowed the ilia to slide anteroposterially, rotate laterally, and rotate dorsoventrally, whereas the sacro-urostylic joint was bicondylar and relatively inflexible (tested via manual manipulation).

## DISCUSSION

In this paper, we have described and characterised the musculoskeletal anatomy of *P. maculatus* for the first time. Paired with traditional dissection, the recently developed DICE μCT technique was used to produce a detailed account of the complex 3D geometry of the distal spine, pelvis, and hindlimb. The work is currently the first digital anatomical description of a terrestrial/arboreal species of frog and represents only the second time DICE μCT has been used in the visualisation and description of anuran musculoskeletal anatomy. The digital dissection conducted here allowed accurate visualisation of muscular anatomy of this species as never seen before. Furthermore, the 3D PDF (Data S1, see

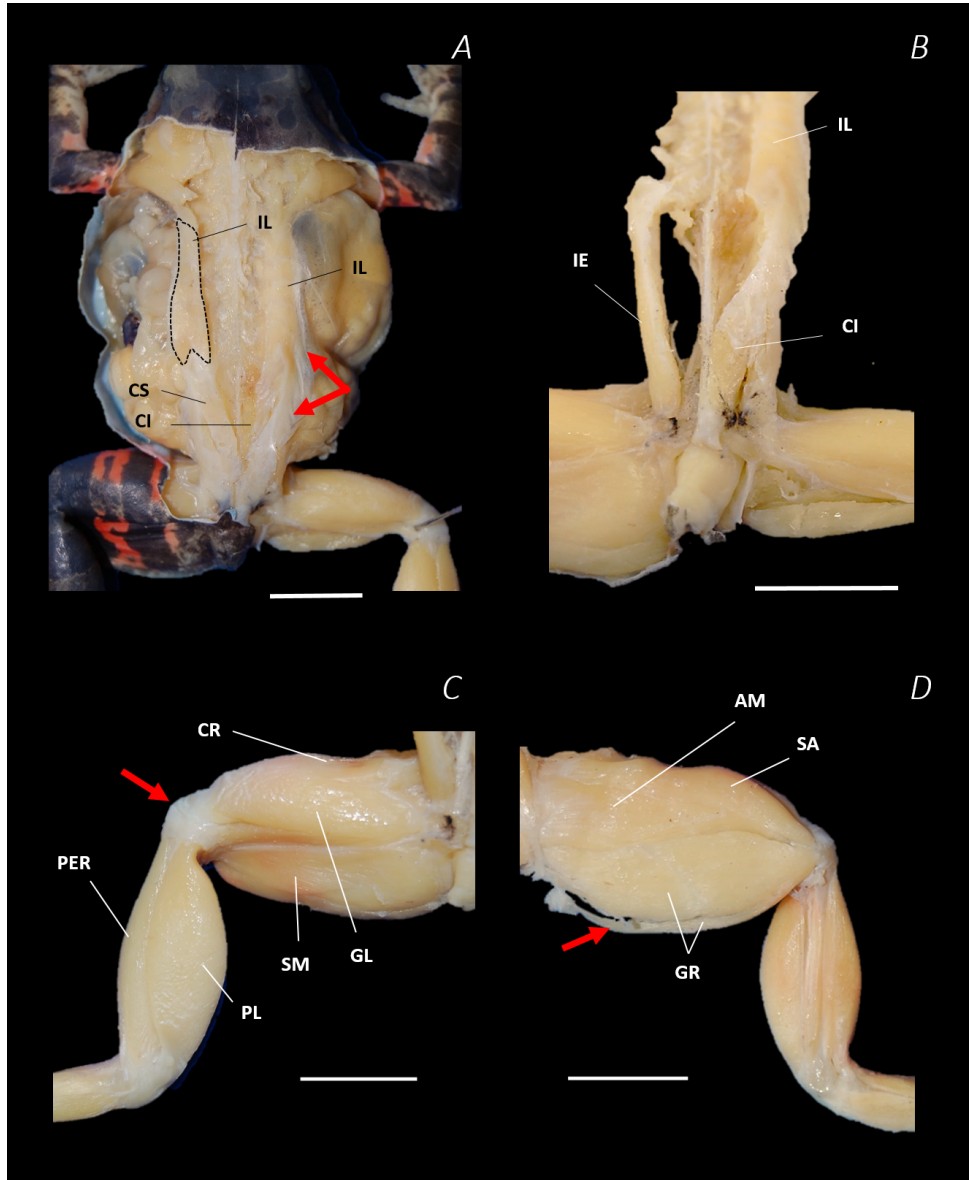

**Figure 7** **Traditional dissection photographs of the dorsal body (A), dorsal pelvis (B), dorsal (C) and ventral (D) left proximal hindlimb.** The red arrows indicate the dorsal fascia in (A), the knee aponeurosis in (C), and the small gracilis minor muscle in (D). The black dashed lines in (A) depict the external borders of the left IL muscle, note the posterior split. Scale bars are shown in white, all of which are 1 cm. See Table 2 for muscle abbreviations.

Article S1 for 3D PDF user guide), allows readers to perform a non-destructive and repeatable digital dissection of this species for themselves.

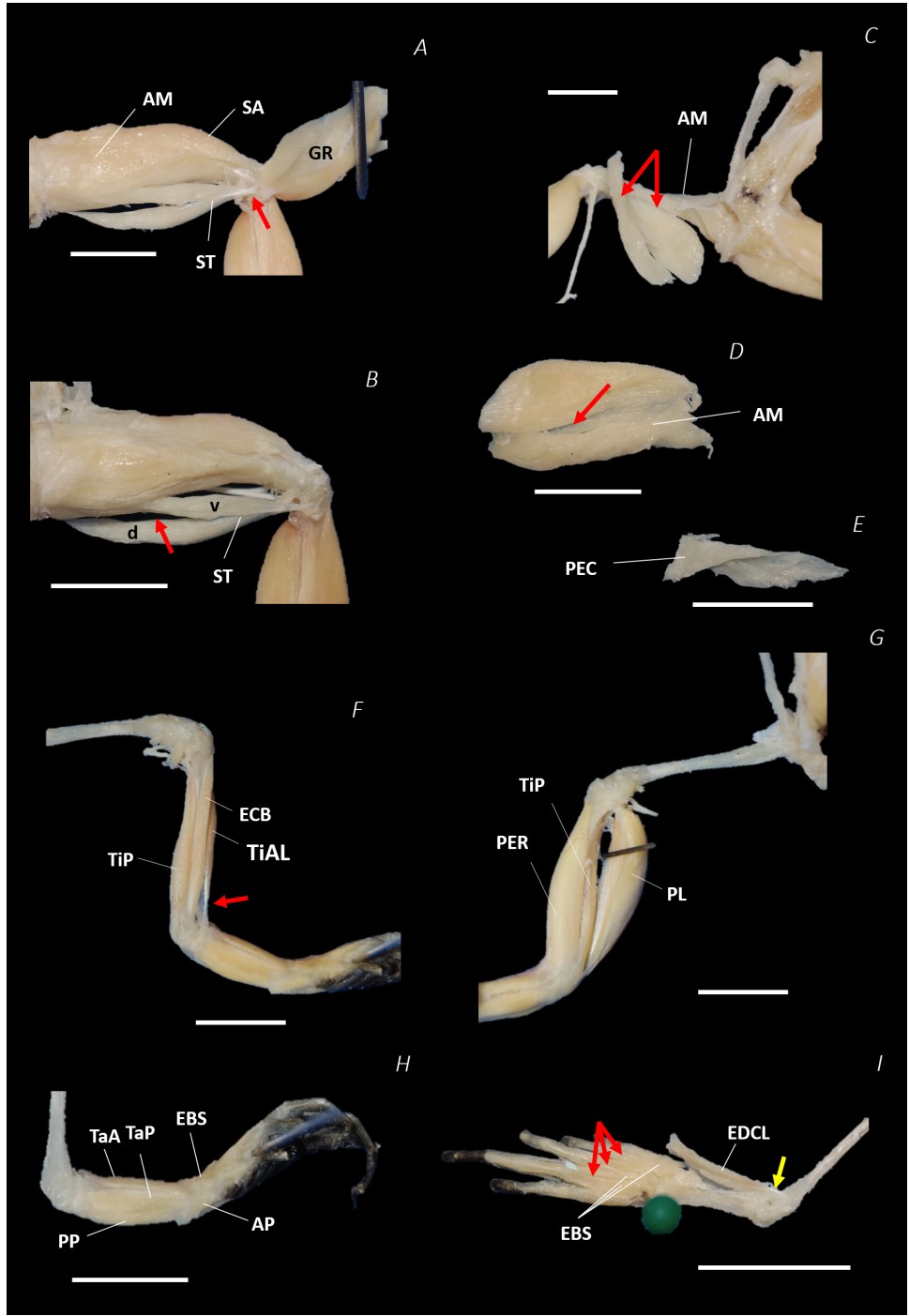

**Figure 8  Traditional dissection photographs of the left femur.** (A) Ventral view of ST tendinous insertion with GR reflected, (B) ventral view of the dorsal (d) and ventral (v) heads of ST with GR removed entirely, (C) posterior oblique dorsal view of the pelvis and left proximal hindlimb - note AM has been partially dissected and reflected in this image, (D) isolated AM muscle, 

**Figure 8 (…continued)**
(E) isolated PEC muscle, (F) ventral view/medial side of the shank with PL removed, (G) dorsal view/lateral side of the shank, (H) lateral view of the tarsals and foot, (I) dorsal view of the tarsals and foot. The red arrows highlight the shared tendinous insertion of the dorsal and ventral heads of the ST muscle in (A), the insertion of STv into the AM muscle belly in (B), the two portions of the AM muscle in (C), the hiatus between the two AM muscle belly portions (through which the ventral tendon of semitendinosus passes) in (D), the tendinous insertion of TiAL (head 2) in (F), and the multiple tendons of the EBS muscle in (I). A yellow arrow highlights the tendinous origin of the EDCL in (I). Scale bars are shown in white, all of which are 1 cm. See Table 2 for muscle abbreviations.

## Musculoskeletal geometry
### The use of DICE µCT scans, 3D PDF, and digital dissection
Dissections revealed intricate musculoskeletal anatomy within the hindlimb and pelvic apparatus, consisting of a large number of muscles, and multiple instances of convoluted curved muscle pathways, where muscles wrapped around bony and soft tissue structures or passed through other muscles. Using DICE µCT enabled us to capture and preserve the 3D topography of the musculoskeletal system of this species in a level of detail that is challenging to achieve using traditional methods. DICE µCT has many other advantages over traditional dissection, including its non-destructive nature, opening this technique up to the anatomical investigation of specimens that cannot undergo destructive sampling. Moreover, iodine staining has been suggested to be reversible (*Bock & Shear, 1972* cited in *Jeffery et al., 2011*), allowing repeated scanning of the same sample with optimised stain concentrations for different features (*Jeffery et al., 2011*).

### Comparative musculoskeletal anatomy
Comparing our dissection findings from *P. maculatus*, with those of other anuran species (*Dunlap, 1960*; *Přikryl et al., 2009*), we found the different anatomical regions exhibited different levels of variation among species. The most variable regions were the spine and pelvis, and the tarsals and proximal foot. Whereas there were fewer examples of anatomical variation in the thigh, and the shank appeared uniform across species.

The anatomical variation of the muscles of the back and the pelvis included variation in insertion sites, some muscle belly shapes, and muscle presence/absence/fusion. While the LD in *P. maculatus* inserted approximately half way down the urostyle, its insertion site in other species ranges from the anterior portion, as in *Pelophylax kl. esculentus*, to the very tip of the urostyle, as in *Ascaphus truei* (*Dunlap, 1960*). The IL insertion in *P. maculatus* is also more proximal than observed in species such as *Kaloula pulchra* where it extends down onto the iliac shaft (*Dunlap, 1960*; *Přikryl et al., 2009*). While the CS was present as a separate muscle in *P. maculatus*, in *X. laevis* and *Barbourula busuagensis* the CS and LD are fused into one muscle belly (*Dunlap, 1960*; *Přikryl et al., 2009*). Compared with *P. maculatus*, which has a clear PY muscle, the PY of *X. laevis* is reduced and further, is altogether absent in *Pipa pipa* (*Dunlap, 1960*; *Přikryl et al., 2009*; *Porro & Richards, 2017*). In contrast, *A. truei* and *Leiopelma hochstetteri* possess a caudopuboischiotibialis (the ancestral trait) not present in any of the other species studied by *Dunlap (1960)* or indeed *P. maculatus*. Finally, the IE muscle of *P. maculatus* was narrow and cylindrical, similar to

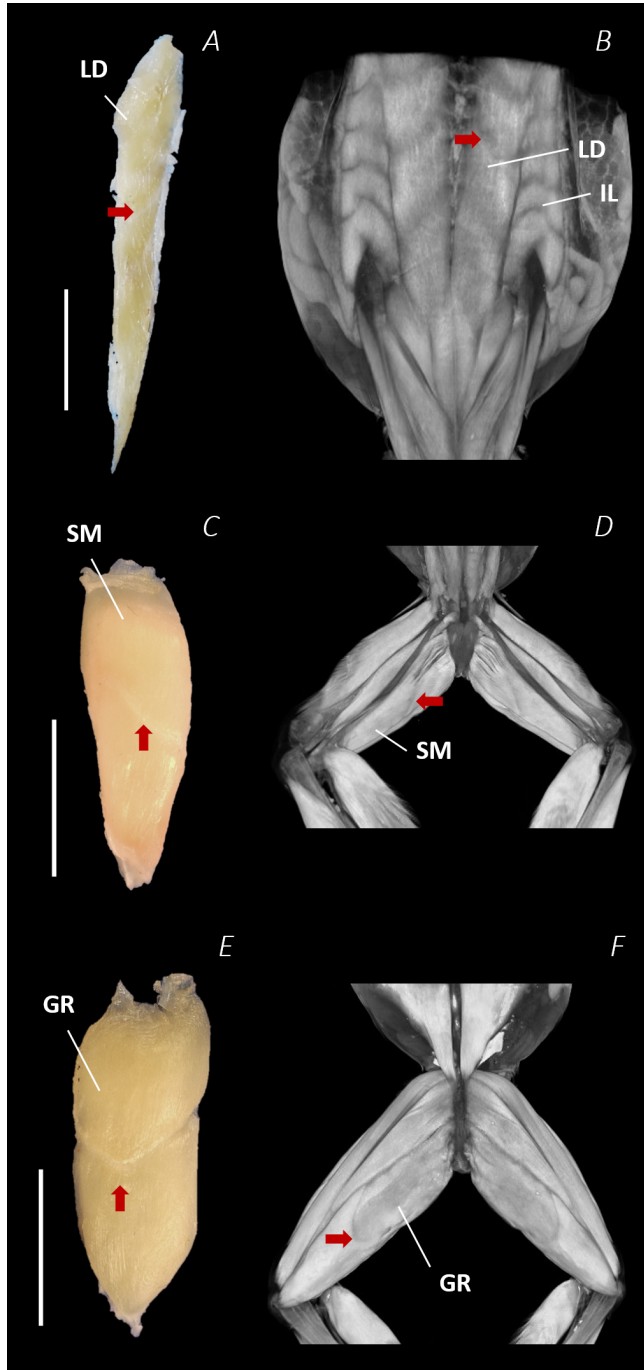

**Figure 9 Isolated dissected longissimus dorsi (A), semimembranosus (C), and gracilis major (E) muscles alongside the reconstructed scan images of the external surface of longissimus dorsi and iliolumbaris (B), semimembranosus (D), and gracilis major (F).** Red arrows are used to show the presence of intersegmental and separating septa. Scale bars are shown in white, all of which are 1 cm. See Table 2 for muscle abbreviations.

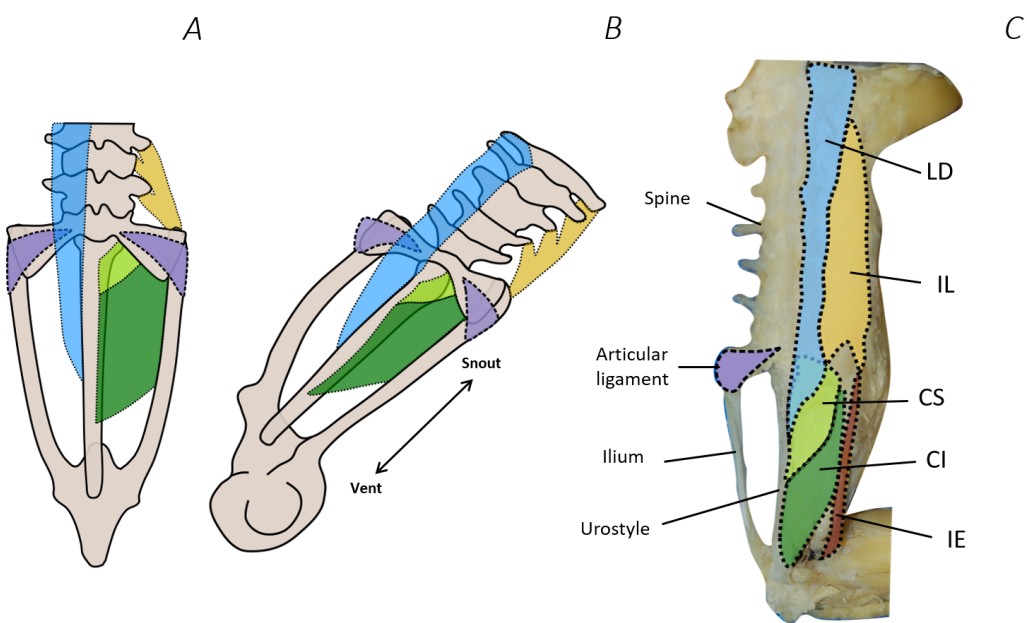

**Figure 10 Comparison between Emerson's characteristic Type IIA pelvic morphotype and traditional dissection data from *Phlyctimantis maculatus*.** (A) and (B) schematic diagrams adapted from *Emerson (1982)* and *Emerson & De Jongh (1980)* show dorsal and posterior-oblique dorsal views, respectively. (C) Shaded traditional dissection photograph of the dorsal spine and pelvis of *P. maculatus*. LD, longissimus dorsi, blue shading; IL, iliolumbaris, yellow shading; CS, coccygeosacralis, light green shading; CI, coccygeoiliacus, dark green shading; IE, iliacus externus, red shading. Articular ligament shaded purple.

*Rhaebo guttatus*. In other species the IE appears broader and more fan-shaped (*Přikryl et al., 2009*), whereas *A. truei* and *L. hochstetteri* it is fused with the II muscle (*Dunlap, 1960*).

The majority of the muscles of the thigh appear uniform among species, however variation is observed in the presence/absence of the AL, number of heads present in the CR and ST, and fusion of some muscle bellies. While *P. maculatus* shares the presence of AL with *Discoglossus pictus*, *B. busuagensis* and *Pelobates fuscus*, the AL is absent in, *Bombina orientalis*, *B. bombina*, *B. pachypus*, *Alytes obstetricans*, *Spea hammondii*, *Megophrys montana*, *X. laevis*, *P. pipa*, *Rhinophrynus dorsalis*, *Atelopus cruciger*, *Polypedates leucomystax*, and *Chiromantis xerampelina* (*Dunlap, 1960*; *Přikryl et al., 2009*). Furthermore, while in *P. maculatus*, and most other species, CR is one muscle belly, in specialised aquatic species, the CR muscle is divided into two incomplete heads (*Dunlap, 1960*). Finally, *A. truei* and *L. hochstetteri* exhibit fusing and combinations of multiple thigh muscles (SA and ST; AL and PEC; QF and GE) that were not observed in *P. maculatus* (*Dunlap, 1960*).

While muscles of the shank were not variable, multiple differences in muscle proportions, attachment sites, presence/absence and splitting/fusion are evident in the tarsals and proximal foot. Like most other species, *P. maculatus* had two distinct heads of the TiAL muscle, which were of roughly equal size. *Dunlap (1960)* reported variation in the proportions of the two heads, and the position at which the muscle bellies of TiAL diverge. The TaP muscle of *P. maculatus* is similar to that of other Ranids but is smaller

than seen in *A. truei* and *L. hochstetteri* (*Dunlap, 1960*). The EBS muscle splits into multiple heads in all species however the digit upon which the middle head of this muscle inserts is variable among species (*Dunlap, 1960*). As in the other areas of the hindlimb, the highly specialised aquatic species, such as *P. pipa* and *X. laevis,* lacked muscles which were present in *P. maculatus*, such as the EBS, AP, and TPP and D. Additionally, while in *P. maculatus*, the PP remains separate from FDBS, in *A. truei* and *L. hochstetteri* these two muscles are fused (*Dunlap, 1960*).

Overall, *P. maculatus* had a similar gross muscular anatomy to *R. catesbeiana* and *R. guttatas* but differed from those species that exhibit ancestral traits (leopelmids) and those that are highly specialised (pipids) (*Dunlap, 1960*; *Přikryl et al., 2009*). The highly specialised, *P. pipa* and *X. laevis* lacked muscles which were present in *P. maculatus*, such as the PY (which is only reduced in *Xenopus* (*Porro et al., 2017*), CI, EBS, AP, and AL. Whereas, *A. truei* and *L. hochstetteri* exhibit fusing and combinations of multiple muscles (SA and ST; AL and PEC; OE; QF and GE; PP and FDBS; IE and II) that were not observed in *P. maculatus* (*Dunlap, 1960*).

### Diversity of hindlimb/pelvis muscle morphology in relation to diversity in function

In the pelvic region, there is strong evidence linking variation frog musculoskeletal anatomy to diversity in locomotor style (*Emerson & De Jongh, 1980*). The region differentiates after metamorphosis and thus the variation seen in this musculature seems to relate to expansion of locomotor capacity as froglets develop (*Fabrezi et al., 2014*). Indeed, differences in the LD, CI, CS, and IL muscles contribute to functional differences in lateral bending and gliding of pelvis associated with walking and swimming, respectively (*Emerson & De Jongh, 1980*). The II, IE, and TFL are derived muscles also likely to influence locomotor mode (*Přikryl et al., 2009*). For example, IE length correlates with jumping (*Fabrezi et al., 2014*), thus may be expected to have a similar morphology among jumpers. However, IE morphology of *P. maculatus* differs from other jumpers (e.g., Ranids) owing to its cylindrical shape (see above). Given that the IE may play an important role in bringing the leg upwards and forwards (*Kargo & Rome, 2002*), perhaps its shape represents a modification to assist with the swing phase of walking/running.

Compared with the pelvis, evidence for interspecific muscle structure-function relationships in the hindlimb is relatively sparse. As stated above, the thigh region appears more conserved compared with the pelvis and distal hindlimb among species, including *P. maculatus*. Despite conservation of thigh muscular traits among modern taxa, *A. truei* and *L. hochstetteri* (representing "primitive" taxa) exhibit fused musculature (see above). In absence of further modelling analysis (e.g., *Kargo & Rome, 2002*), we can only speculate on the functional consequence of fused muscles. Perhaps separate versus fused muscles allows for a greater variation in muscle moment arms, thus increasing the functional workspace of the limb hence enhancing the limb's ability to perform multiple tasks.

Apart from the above discussion based on a small number of species and focused anatomical regions, we are far from a clear and complete understanding of how interspecific variation in muscle morphology relates to locomotor function. In light of broad patterns

relating limb, body, and skeletal morphology to locomotor style (*Zug, 1978*; *Emerson, 1988*) as well as performance, ecology, and phylogeny (*Moen, Irschick & Wiens, 2013*), we also expect relationships to emerge with muscular morphology. Although certain features of the musculoskeletal system seem to vary independently of ecology/function (*Fratani, Ponssa & Abdala, 2018*), we expect that other traits may exhibit morphologically small differences amounting to profound effects on locomotion. In particular, small changes in muscle moment arm distances have a strong impact on muscle function (*Lombard & Abbott, 1907*; *Kargo & Rome, 2002*). For example, there is great diversity in subtle aspects of bone shape (e.g., ridges on ilia; *Reilly & Jorgensen, 2011*) which may alter muscle origins/insertions significantly enough to change muscle moment arms. Moreover, muscles are not mechanically independent; the function of a single muscle depends on the current configuration of the joints, thus is dependent on the action of other muscles (*Lombard & Abbott, 1907*; *Kuo, 2001*; *Kargo & Rome, 2002*). Consequently, analyses that treat all muscles independently might overlook synergistic effects of small changes in one muscle with respect to other muscles. Regardless, the species currently sampled for detailed muscle analysis are likely too few to disentangle function/ecology from phylogenetic effects. Fortunately, recent workers have assembled a vast archive of frog digital anatomy (open Vertebrate; "oVert") and are currently making it available for exploration and study (*Watkins-Colwell et al., 2018*). Thus, future work can build upon our observations as well as past work to accumulate sufficient intraspecific data to apply rigorous comparative/phylogenetic methods (e.g., *Moen, Irschick & Wiens, 2013*; *Fratani, Ponssa & Abdala, 2018*) towards elucidating muscle morphology-function relationships across Anura.

### Gracilis major and Semimembranosus myology

The split found in the SM and GR (major) is unique among the dozens of muscles of the leg and pelvis. Although this tendinous "septation" is well-characterised in all other anurans observed (*Dunlap, 1960*; *Duellman & Trueb, 1986*; *Přikryl et al., 2009*), neither the developmental mechanism nor the biomechanical significance of this muscle structure is known. One possible explanation is incomplete fusion of two evolutionary/developmentally distinct muscles which appears as intramuscular separation. However, *Přikryl et al. (2009)* note that the hindlimb extensor muscles in modern frogs have not changed over the course of evolution; thus, we speculate that separation may not be a developmental/evolutionary artefact, but rather may have functional significance. For example, tendinous separations between muscle bellies are not unique to anurans; separated muscles have been described in salamanders (*Ashley-Ross, 1992*; *Walthall & Ashley-Ross, 2006*) and cats (*Bodine et al., 1982*). In those cases, the divided muscle bellies receive separate innervation (*Francis, 1934*; *Bodine et al., 1982*) suggesting that two regions of a single muscle could act independently. If the partitions of the SM and GR were found to have separate innervation, we propose a potential mechanical function as follows. Both the SM and GR muscles span from the pelvis to the knee and insert into the aponeurosis covering the knee. The SM in particular has been recorded to function both as a femur retractor and a knee flexor and, while it does not act on the knee joint, GR is also dual functional, acting to either retract or adduct the

femur (*Přikryl et al., 2009*). We speculate a separation could act to partition the portions of muscle designated for each role, allowing the animals to fine tune hindlimb motion. Further anatomical investigation determining innervation and/or spatial partitioning of fibre types would be required to better characterise these muscles. Subsequently, modelling analyses could be performed to assess the biomechanical impact of intramuscular separation.

### *Phlyctimantis maculatus pelvic morphotype*

Given recent findings that *P. maculatus* are walkers/runners as well as hoppers, jumpers, and climbers (*Porro et al., 2017*), we evaluated whether the pelvic morphohotype represented walker or jumper traits. Pelvic dissection revealed that *P. maculatus*'s anatomy was consistent with the Type IIA morphotype defined by *Emerson (1979)*. The sacrum was dorsoventrally flattened as opposed to the cylindrical sacral shape of Type IIB jumping species, yet lacked the extreme laterally flared diapophyseal expansion and broad ligamentous cuff of the Type I pelvis, as seen in aquatic, swimming species such as *P. pipa* (*Emerson, 1979*; *Emerson, 1982*; *Přikryl et al., 2009*).

However, some of the key features of the Type IIA morphotype were subtler in *P. maculatus* compared with other walking and hopping, and burrowing species. For example, *R. guttatus, B. busuagensis, D. pictus S. hammondii, Rentapia hosii, A. obstetricans, Anaxyrus (Bufo) americanus, A. boreas* are walking, hopping, or burrowing species that exhibit broader lateral flaring of the sacral diapophysis and a more obvious bow-tie shape sacrum (*Přikryl et al., 2009*; *Reilly & Jorgensen, 2011*). Although dorsoventrally flattened, the sacrum of *P. maculatus* demonstrated less flaring and more posterior lateral projection of the sacral diapophyses, similar to the skeletal morphology of *A. truei* and *A. montanus* (*Reilly & Jorgensen, 2011*). Perhaps the typical Type IIA morphotype features are less prominent in *P. maculatus* because of its more arboreal ecology and use of both walking and jumping behaviour (as opposed to hopping).

Furthermore, *P. maculatus* also exhibited slightly dorsally ridged ilia and urostyle. Having prominent ridges along the dorsal surface of the ilia is a trait commonly seen among the sagittal-hinge Type IIB morphotypes (*Emerson, 1979*; *Emerson, 1982*; *Reilly & Jorgensen, 2011*). Nonetheless, *Reilly & Jorgensen (2011)* reported the same set of features in other representatives of Hyperoliidae (*Kassina senegalensis* and *Hyperolius lateralis*), summarising that the pelvic girdle design of this family comprises an expanded sacrum, iliac ridges, and half urostyle ridge. They also categorised this family as walkers, hoppers, and arboreal jumpers. *P. maculatus* is therefore consistent with their description of hyperoliid frogs. Furthermore, manual manipulation of the ilio-sacral joint prior to and following muscle dissection suggested a capacity for lateral rotation, some anteroposterior sliding, and sagittal bending. Since lateral rotation of this joint is only freely permitted by the Type IIA morphotype (by definition *Emerson, 1979*; *Emerson, 1982*), we felt justified in classifying the pelvis of *P. maculatus* as such. It should also be noted that while the degree of sacral diapophyseal expansion is a useful trait in distinguishing between the sagittal hinge or lateral bender morphotype, the extent of sacral diapophysis lateral expansion is highly variable among anurans (*Jorgensen & Reilly, 2013*).

Given the extent of subtle variation observed in the musculoskeletal anatomy of anurans in general, it seems likely that Emerson's categories represent the archetypical morphotypes within each behavioural group (walking/hopping, jumping, and swimming). Those species that use multiple locomotor behaviours possess a subtle blend of pelvic characteristics. Thus, rather than fitting discrete morphotypes, frogs more likely span a continuum of pelvic morphologies depending on the combination of locomotor behaviour expressed by a given species, as suggested previously (*Soliz, Tulli & Abdala, 2016*).

### Limitations and caveats of DICE μCT

DICE μCT has proved an excellent method to present the 3D topography of the musculoskeletal system, allowing the visualisation of complex 3D interactions between muscles and other structures not possible with traditional techniques. However, there are some notable caveats associated with DICE μCT that should be discussed here.

The aim of the DICE μCT technique is for the stain to disperse through the soft tissue, increasing contrast (Figs. 2A–2C), as such, poor diffusion or too low a stain concentration results in poor contrast enhancement. Here, we skinned the specimen in order to assist with stain perfusion however a lack of published methodologies for amphibian staining means we cannot comment on how effective removal of the skin was for this purpose. On the other hand, poor soft tissue contrast can also result from overstaining. Furthermore, a fine balance needs to be struck in order to avoid distortion of the sample due to extreme tissue shrinkage.

Tissue shrinkage is a commonly reported caveat of DICE μCT (*Vickerton, Jarvis & Jeffery, 2013*; *Cox & Faulkes, 2014*; *Buytaert et al., 2014*; *Gignac et al., 2016*; *Bribiesca Contreras & Sellers, 2017*). Muscle volume shrinkage due to contrast enhanced staining has been previously reported ranging from 10–56% (*Vickerton, Jarvis & Jeffery, 2013*; *Buytaert et al., 2014*; *Bribiesca Contreras & Sellers, 2017*). The extent to which shrinking occurs in stained specimens increases with higher concentrations of $I_2KI$ and can be reduced by using lower concentrations over a longer duration as was implemented in this study (*Vickerton, Jarvis & Jeffery, 2013*; *Gignac & Kley, 2014*). Contrast enhanced staining is therefore a time consuming process and despite us using low concentrations of Lugol's solution here to avoid it, shrinkage of the muscle tissue was observed in our frog. We do not believe the level of shrinkage in our specimen to be detrimental to the overall results of our anatomical assessment since origins, insertions, and pathways are unlikely to have been affected significantly (*Cox & Faulkes, 2014*). While muscle volumes are not the focus here, it should be noted that measuring the specific level of shrinkage is an important consideration in those studies reporting quantitative muscle data (for example, see *Bribiesca Contreras & Sellers, 2017* for their comparison of dissected vs stained muscle volumes).

A further limitation associated with the DICE methods used here is the inability to visualise tendinous or ligamentous tissues. Important structures such as the knee aponeurosis, muscle tendons, or plantar fasciae were therefore indistinguishable in the scan data and excluded from the 3D PDF. This caveat was also encountered by *Porro & Richards (2017)* who suggest using agents that bind to collagen as an alternative. *Descamps et al. (2014)* demonstrate that phosphotungstic acid (PTA) has a preference for binding to

collagen and connective tissues whereas phosphomolybdenic acid (PMA) provides good contrast for the visualisation of cartilage using CT. When using any chemicals health and safety precautions must be adhered to, not all laboratory spaces are suitable to conduct the aforementioned procedures.

Finally, obtaining the highly detailed results of DICE μCT is time consuming, making analysis of several specimens of one species impractical. Consequently, studies such as this assume low intra-species variation. Moreover, the subsequent segmentation of the μCT scan data requires anatomical expertise and relies on the user's discretion to appropriately define voxel material. The final 3D model is therefore best defined as a 3D representation of the anatomy of an example specimen.

Despite the limitations, DICE μCT as a technique has already begun to revolutionise the way anatomy is visualised and studied. With further use in a wider range of taxa, the protocols are likely to improve and become more standardised as a wider knowledge base for troubleshooting is generated. Even though this is one of the earliest uses of the technique in anurans, the results obtained in this study were remarkable and facilitated a deeper understanding of the gross anatomy of *P. maculatus*.

### Future work

Recently, anatomical reconstructions created using DICE μCT have been used as the foundation for musculoskeletal models to investigate structure-function relationships of the locomotor musculoskeletal system (*Charles et al., 2016a*; *Charles et al., 2016b*; *Allen et al., 2017*). We propose to use our 3D digital model to generate an anatomically accurate musculoskeletal model of *P. maculatus,* allowing us to explore the mechanical effect of the complex curved muscle trajectories, and test our speculative hypotheses regarding the implications of separate innervation in the same muscle belly (such as seen in the SM and GR). Furthermore, our musculoskeletal model can be applied to questions regarding evolutionary adaptations. For example, by altering muscle attachment sites, and/or skeletal proportions to mimic those of extinct species, we can explore the functional significance of such adaptations.

## CONCLUSIONS

Here we present a complete assessment of the musculoskeletal anatomy and 3D geometry of the lower spine, pelvis, and hindlimb of *Phlyctimantis maculatus*. Traditional and digital dissection revealed that the musculoskeletal anatomy of *P. maculatus* is comparable to other derived species and, as predicted, their pelvic morphology is consistent with the Type IIA morphotype associated with walking and hopping. The DICE μCT technique was a valuable addition to our methodology, allowing us to visualise muscle interactions in 3D. However, we found this technique still needs to be combined with traditional dissection in order to observe tendinous attachment points. Nonetheless, the anatomical data presented here act as an excellent educational resource and form a crucial platform for further functional analysis within the anuran pelvis and hindlimb. Both the digital and traditional dissections performed are critical for the creation of an anatomically accurate musculoskeletal models that could be used to perform moment arm analyses. Future work

will use such models to investigate muscle function during both walking and jumping locomotion in *P. maculatus*.

## ACKNOWLEDGEMENTS

We thank Mark Hopkinson for invaluable assistance with the μCT scanning and reconstruction. Dr Chris Basu provided valuable comments on draft versions and Dr Zoë Davies contributed interesting discussion of muscle anatomy. Dr James Charles, along with Dr Sandy Kawano, further assisted in the creation of the 3D PDF. Thank you to all reviewers for their thoughtful comments and suggestions.

### Funding

This work was funded by a European Research Council Starting Grant (PIPA338271) awarded to Dr Christopher Richards. The funders had no role in study design, data collection and analysis, decision to publish, or preparation of the manuscript.

### Grant Disclosures

The following grant information was disclosed by the authors:
European Research Council Starting Grant: PIPA338271.

### Competing Interests

The authors declare there are no competing interests.

### Author Contributions

- Amber J. Collings conceived and designed the experiments, performed the experiments, analyzed the data, contributed reagents/materials/analysis tools, prepared figures and/or tables, authored or reviewed drafts of the paper, approved the final draft.
- Christopher T. Richards contributed reagents/materials/analysis tools, authored or reviewed drafts of the paper, approved the final draft.

### Animal Ethics

The following information was supplied relating to ethical approvals (i.e., approving body and any reference numbers):

The UK Home Office provided full approval for all research conducted on this grant (License 70/8242).

### Data Availability

The raw data is available at MorphoSource: https://www.morphosource.org/Detail/SpecimenDetail/Show/specimen_id/18771.

### Supplemental Information

Supplemental information for this article can be found online at http://dx.doi.org/10.7717/peerj.7003#supplemental-information.

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
