# Peer review of "Digital dissection of the pelvis and hindlimb of the red-legged running frog, Phlyctimantis maculatus, using Diffusible Iodine Contrast Enhanced computed microtomography (DICE μCT)"

_PeerJ, doi:10.7717/peerj.7003_

## Round 0.1 · original submission · Major Revisions

I am sorry for the delayed response. I have now received three reviews of your manuscript. All of them agree in that your study presents interesting data. They also give you many suggestions that will improve your work considerably. I will summarize some of the points that need your attention:

All reviewers detected some problems with your figures, which are a major contribution of your work. Please pay attention to these details. Reviewer 1 ask for clarification of the difference in staining between rounds three and four. It is important to make more precise your identification of the tendinous structures. Reviewer 2 and 3 have problems in relation to the categories related to locomotor modes, please check them, especially the generalist one. They also agree in that a general reorganization of the manuscript is needed. I expect you to take all reviewers’ recommendations in full consideration.

Reviewer 1 ·

Basic reporting

The study presents a detailed description of the lower back, pelvic girdle and hindlimb elements of the musculoskeletal system of Kassina maculata. Hypotheses and results are relevant and the manuscript is overall well written in clear and unambiguous English. I have a few suggestions to improve the manuscript which are detailed below.

Kassina maculata was transferred to Phlyctimantis in the phylogeny of Portik and Balckburn (2017 Evolution 70-9: 2017–2032). Please clarify why you choose not to adopt the latest systematic proposal. 

In the introduction it would be nice to expand a little on the morphological adaptations of K. maculata which enabled the occupation of terrestrial and arboreal habbitats (lines 59-61).

The 3D digital models are very good, however most figures are lacking the scale bar. More importantly, figures showing gross anatomy can be improved. Most lack the scale reference, there is variation on background color (figure 8) and position of labels (lettering). Figure 7B has very poor quality. The spaces between each figure of fig 9 are visibly different, and it would be more elegant to use a scale bar on figures 9D and E. There is no need of adding a red arrow to highlight an element if it already has a legend (iliacus externus in fig 7 and plantaris longus in fig 9). Also, please check the order of reference of figures in the text.

Experimental design

The manuscript presents original research, with a well-defined and relevant research question, detailed methodological description and high technical standard.

Regarding the methods section, Table 1 is very informative, but I was unable to understand the difference in staining between rounds three and four, besides resolution. The authors recommend further staining on round 3, and there was no modification neither in stain concentration nor in time. Please clarify.

Validity of the findings

In the results, the anatomical description of muscles includes the tendinous elements, which is great! However, I noticed some important ones were missing. For example, the plantaris longus usually has two tendinous proximal attachments and inserts by the Achilles tendon, and those were not mentioned. The longissimus dorsi may have a tendinous insertion attaching to the urostyle, and the iliacus externus usually attaches to the proximal end of the femur by a strong tendon. Please check.

The discussion is well stated and linked to the research question. I suggest the authors take a look also into Fabrezi et al. (2014 Evol Biol DOI 10.1007/s11692-014-9270-y) and Fratani et al. (2018 Zoology, 126, 172-184) which include information on muscle and tendon variation and its relation to locomotion.

Additional comments

Minor comments:

62 – “robust to maintain” please rephrase that. Maybe add “in laboratory conditions”?

79 – While Xenopus is regularly used “as” a model…

145 – You should use “specimen” instead of species here. Please provide collection numbers for the individuals used in the study.

162-164 – This sentence should go in the materials & methods section.

176 – Does the Longissimus dorsi has a tendinous insertion in the urostyle? Not clear in the figures.

240 – (…) K. maculata “does” not deviate…

363 – (…) in “a” detail “level” that is challenging...

384-386 – Please include a reference paper for this sentence.

437 – (…) biomechanical significance of this “muscle pattern”...

488 – I don’t understand “exemplar specimen” here. Did you mean example specimen? Single specimen? Model specimen?

516 – (…) are critical for “the” creation of anatomical…

Table 2 - Extensor brevis superficialis (EBS) and flexor digitorum brevis superficialis (FDBS) appear to be missing.

Figure captions:

Figure 6 – There is no muscles in this fig so the sentence “See main text, or Table 2 for muscle
abbreviations” is not necessary.

Figure 9 – Please clarify in each caption if it is dorsal or ventral view of the specimen. There is a point where it is supposed to be a comma after semitendinosus.

Reviewer 2 ·

Basic reporting

Overall the paper is well written in clear English, however, I think the writing still needs some polishing throughout the manuscript. I have pointed out at some examples below. Although the anatomical descriptions are clear and used terms are consistent with the nomenclature, they are too long so I would recommend summarizing the anatomy section of the Results. It might be important to consider how relevant these are, as muscle positions are summarized in Table 2. Maybe adding an extra column to Table 2 with ‘Muscle description’ and giving a short description such as: ‘thin parallel muscle’ would help to reduce the descriptions in Results. Often, the muscles are compared to other frog species, which I consider should be part of the discussion instead (see below).
Scientific names are used throughout the manuscript, however, according to taxonomic nomenclature it is expected to abbreviate the scientific names after the first time a species name is mentioned and continue abbreviating from then on. So, Kassina maculata, should become K. maculata the second time it is mentioned. As well as the species genus, if the genus Rana is mentioned and later on the text a different species from the same genus is included it should be abbreviated as R. spp. The literature is well referenced, relevant and extensive. A few points are highlighted below, where I consider necessary to check the literature again and extend it. The segmentation and the 3D model are very well done, some comments for the figures are explained below that have to be addresed, a scale bar is needed in most of the images.

Experimental design

This paper investigates the gross anatomy of the musculoskeletal system of the spine, pelvis and hindlimb of K. maculata, as a generalist model (in locomotion modes) of a frog using diceμCT and gross dissection. The authors stated the aims clearly, although some re-writing should be done (see below), and present the first 3D model of the spine, pelvis and hindlimb anatomy of a frog with multiple locomotion modes that has potential use in a number of studies of functional morphology and biomechanics of jumping and walking. The methods are well defined but the writing needs some polishing. I would recommend to summarise Table 1, and include in Methods the staining and scanning details used for the final scan from where the model was generated (see below).

Validity of the findings

The research undertaken in this work is meaningful, the results are clear and the conclusion is relevant and well stated. The addition of a 3D PDF available is great and provides a useful tool for anatomical studies of anurans. The findings are interesting and deserve publication, once the comments included here are addressed.

Additional comments

Introduction
Line 59. It should read ‘red-legged running frog…’
Line 63. Generalist of what? Locomotion modes?
Line 74. Literature is incomplete, a number of papers have provided tools to visualise soft tissue anatomy of different animal groups through diceμCT. See, for example, Bribiesca-Contreras and Sellers 2017 and Düring et al. 2013.
Lines 82-88. Aims of the work should be re-written as it is hard to follow in its current form. Capital letters do not follow after commas, it should be changed to lower case.
Line 87. Specify the source and location of the library and groups that are represented on it. It is not clear from the text.

Materials & Methods
Line 91-92. Which one is the subheading? If both, one should be highlight some other way to distinguish between the two.
Line 95. Is the removal of the heart part of the staining procedure? It should be explained why the heart was removed.
Line 96-97. Sentence is not clear: ‘to limit internal exposure to fixative and, initially, staining solution’. What is stained first and then fixed? The following sentence states that the sample was fixed first. The sentence should be re-written.
Line 99. Specify if this was done by moving the specimen to a PBS solution where if was left soaking overnight or by rising the specimen or so. It is not clear.
Line 100. ‘using 1 x phosphate buffered saline (PBS)…’ could be changed to ‘using a PBS solution’.
Line 101. ‘To enhance contrast’, delete ‘the’.
Line 102. What is the concentration of the solution? Include in the text or if different concentrations were used for each round of staining and scanning specify so or refer to table 1.
Line 103- 105. ‘The specimen underwent five rounds of staining…’ This sentence is too long and not too clear. It should be re-rephrased.
Line 138. Define acronym TMT in first use.
Line 146. Re-phrase sentence ‘This animal had been euthanized following experimental data collection not discussed here (euthanasia methods as above’. Perhaps something like ‘This animal had been euthanized as mentioned above’?

Results
It is not recommended to have references in results, although, is tempting in anatomical studies like this. Keep them to the minimum if you consider it extremely important to be mentioned here or, I recommend moving the comparisons with other species (lines 177, 179-180, for example) to the discussion.

Line158. A comma is missing after digits.
Line 162. Just saying ‘Figure 3-6’ is enough.
Line 162-163. ‘All muscles are named…’ This sentence is weird and should be re-phrased. Is there any nomenclature you used in particular? The muscle names are consistent in all these papers?
Line 210. Delete ‘for PY position’ in the brackets.
Line 214-216. This sentence can be moved to discussion and start the paragraph with something like: ‘In K. maculata the II originates…’.
Line 218-219. ‘Both the TFL and GL insert on…’.
Line 227. Add a comma before ‘however’.
Line 299. ‘…form the majority of the muscle tissue…’, do you mean mass? If so, it should be specified. The whole sentence should be re-phrased as is hard to follow.
Line 245-248. This sentence should go in the Discussion.
Line 263. ‘The smaller portion has similar origin and insertion…’.
Line 268. ‘Gemellus’ should start with lower case.
Line 268-269. The sentence within the brackets does not follow any previous idea, it should be deleted or moved as part of the discussion.
Line 325. ‘Origination’ should be ‘origin’.
Line 330-334. This sentence should be moved to the Introduction and, if necessary, mention a brief summary in the Results. The following sentence should be re-worded as it can’t be followed easily.

Discussion
Discussion is clear and consistent with the aims. The limitations of the method employed are well stated and clear. I would recommend however, including more references when talking about the limitations of using diceCT as multiple studies have discussed issues with staining, such as tissue shrinkage. It would be also important to add what you did to resolve those issues. Explaining why you decided to skin the sample during the staining period could help others to replicate your method. The anatomy discussion is good and clear, moving some parts from results could make it easier to understand (see above).

Line 349. Delete the comma after ‘first’.
Line 353-355. This sentence needs to be re-written.
Line 384. Reference is missing.
Line 400. Move reference to the end of the sentence in Line 402.
Line 466. A reference is missing at the end of the sentence, if the entire idea comes from Vickerton et al. 2013 this reference should be moved at the end of the sentence. Shrinkage of tissue has been studied repeatedly in similar studies using diceμCT, examples could help to make your point stronger as levels of shrinkage are not reported in this study. For studies where quantitative data of muscle is needed, knowing the shrinkage level would be very useful.

Conclusions
You do not mention anything about the use of diceCT in your conclusions. This is a crucial part of your study and such, it should be mentioned here. A brief conclusion stating in the technique was useful or not to visualise the soft tissue anatomy of the pelvis and hindlimb of K. maculata, and what is the bigger issue with this, such as the lack of tendons and mention that this is why it is important to combine traditional dissection.

Line 507. I don’t believe you are providing any quantitative data in your study.
Line 509. It is not common to include references in the conclusion. Re-phrasing the sentence could help to remove them or, at least, make the idea clearer. ‘According to previous morphological descriptions, K. maculata possess… ‘.

Figures and tables.
Figure 1. Add scale bar. The quality of the figure could be improved, traces of cellophane are evident in B.
Figure 2. Again, a scale bar is needed and quality could be improved. Adding a black background to the image would make it cleaner and the divisions of the different views won’t be evident.
Figure 3. I don’t understand what you mean by surface, it is superficial? ‘Superficial digital dissection’ could be best. Re-phase second part: ‘See main text and Table 2 for muscle abbreviations. For interactive 3D PDF, see Supplemental Information’.
Figure 3-6. Figure legends must be improved, see above. The axis is barely visible in all the images, its size must be increased and a scale bar must be included.
Figure 7-9. The figure legends should be polished. For example, in Fig. 7., you can write something like this: The red arrow indicates the dorsal fascia in A, the fleshy iliacus externus […] in B and […] in C. The dashed lines show…’. In Fig. 8, you can write: ‘** show the presence of intersegmental septa and * denotes the separating septa’.
Table 1. I think this table is long and some parts should be better summarized, you don’t need two columns for the number of rounds and name of scan test type. Also, it could help the readers to include in Methods what parameters where changed during the scan test, such as solution concentration and skinning of the specimen, and mention why you decide this. As well as mentioning the details for the final scan used to produce the model.

Reviewer 3 ·

Basic reporting

The manuscript presents a very interesting and useful methodology for the study of the musculoskeletal system in Kassina maculata, a specie with multiple locomotor modes and habitat uses. The technique is well explained and is useful as a guide for the detailed study of the musculature of this and other species. Although it is a non-destructive technique that allows keeping the material intact, it not allows visualizing tendons and ligaments. For that reason, the authors also carried out traditional dissections, which together allow a more detailed study of the musculature and the pelvic morphology.

The manuscript is clear, unambiguous, and there is a professional English language used throughout. Anyway, I suggest some changes for better reading and comprehension.

-In the first place, I suggest adding more information about the importance of studied the musculoskeletal system in anurans and the interspecific difference of the musculature.

-Secondly, the authors remark (Line 59) that K. maculata has multiple locomotor modes,[but they don’t add to the list that the specie can also climb (Porro et al., 2017)] and also mentioned that it is a terrestrial and arboreal frog. These make K. maculata a good “generalist frog model system”. Anyway, in line 81 the authors wrote: “We, therefore, present the first detailed digital dissection of a generalist (walking and jumping) terrestrial species”. I wonder why here the "generalist" definition is only limited to walking and jumping. Then, in the discussion section, the authors discuss that K. maculata is also an arboreal frog. I suggest writing a more clear definition of a “generalist frog model system”.

Literature is well referenced and relevant. Anyway, some of the quotes of the manuscript are not ordered chronologically and some of them have not the same format (for example &/and). Also, please check that all references have the same format (for example uppercase or lower case in the titles).

Figures and tables are relevant, well labeled and described. Nevertheless, I suggest some changes to improve them that are detailed below in general comments.
Supplementary material represents a very interactive way to study the muscle. If its possible I suggest editing the name of the viewpoints (for example: instead of back and front, ventral and dorsal views).

Experimental design

This is one of the first studies in anurans to use this very modern and not-destructive technique for the study of the musculoskeletal system. The selected specie has multiple locomotor modes and also is a terrestrial and arboreal specie. This makes the study very interesting not only from the field of comparative anatomy but also for biomechanical and morpho-functional studies
The methods are well detailed and are replicable.
The aims of the studied are well defined, anyway, the authors repeat this information in Line 62 and -a more complete idea- in line 83. I suggest reorganizing the paragraphs and emphasize the three principal goals of the work only once.

Validity of the findings

The data of digital dissection present here is a novelty and useful to extrapolate to other anurans species with similar locomotor modes and habitat use.
Also, the comparative analysis of the musculature with other species is adequate. Anyway, for better comprehension, I recommend checking the organization of the results. It is important that the description of the muscle follow the same sequence of the figures, which is also the sequence of a traditional dissection (from the most superficial layer to the internal layer).
The conclusion is well stated, linked to the research question. Anyway, in line 507 authors wrote that they presented a complete and quantitative assessment of the musculoskeletal anatomy. Instead, I think this research has a qualitative approach.

Additional comments

-Specific observations and recommendations.
*Abstract:
Line 17: I recommend adding the three goals of the work.
In Line 38, the authors remark the importance of the digital dissection of a terrestrial frog, nevertheless they mentioned that these frogs are not only terrestrial but also arboreal. This makes it more interesting for study. I think it would be important to remark this feature here and all over the manuscript.

*Introduction:
Line 46: the quote of Calow & Alexander is incomplete.
Line 47: Please check the paper year of Nauwelaerts & Aerts.

*Mat&Met:
Line 151: lack the quote of Duellman & Trueb 1986

*Results:
Line 154 to 264: I suggest reviewing the first paragraph; it seems that much of the information here is more adequate to material & methods section.
Line 170: It is not necessary to quoted Ecker 1889 in this section. You can add the quote in material & methods.
Line 190: please mention the muscle Coccygeosacralis before to the name abbreviation.
Line 191: The CS and CI can be first seen in figure 3 (which shows the more superficial layer of muscle). The same for IE in line 202. For a better reading please mention since the first figure where the muscle appears. On the contrary, the reader could assume that this muscle appears in the Medial digital dissection.
Line 226, 243: It is better to start the sentence with the complete name of the specie.
Line 288: For a better reading of the text and figures, I suggest to inform that, both TiAL1 and TiAL2 can be already seen in figure 3.
Lines 295 to 300: neither the EBS nor ABP were mentioned in these lines.
Line 306: delete )
Line 322: check the word tarsometatarsal
Line 329: I recommend check the last part of the result section. Most of the information here could be incorporated into the Introduction or Discussion sections.
Line 336: Delete A-C from “Figure 10 A-C”
Line 342: check the word sacro.

*Discussion:
Line 373: Please review in this line and all over the manuscript that Lithobates catesbeianus is now Rana catesbeiana. Also Bufo guttatus is now Rhaebo guttatus. I recommend checking the current names of all the species in http://research.amnh.org/vz/herpetology/amphibia/

Line 375/376. It is not necessary to repeat this information due it has been written in Mat&met.
Line 378-384: I think that if you also discuss the locomotor modes that these species have (all together with the similarity and differences in the musculature) could improve this section. Moreover, it would be interesting to discuss the function of the muscles that are present in Ascaphus truei and Leiopelma hochstetteri. Can it be associated with the locomotor mode or use of habitat?
Line 393. Please mention all the locomotor modes for the specie.
Line 419. In the sentence: “Since lateral rotation of the IS joint is only freely permitted by the Type IIA morphotype (by definition)…” please write the corresponding quote.
Line 433. I think this section “GR and SM myology” can be incorporated into “comparative musculoskeletal anatomy” section. Please write the complete name of the muscle in the subheading.
Line 491 to 496: I suggest incorporate this paragraph to the previous section.
Line 503. I suggest mention GR and SM in this line.
Line 504. I don´t understand the authors when said: “augment anatomy…”
Line 569. The reference is incomplete.
*Figures
Figure 3 to 6: delete or increase the size of the x,y, z-axes in the bottom of the figures.
Figure 6 C. Please check this. Tarsals bones are between tibiale and fibulare and metatarsals.
Figure 7: (B) I suggest improving this photography. The quality is not good enough. (C) the photography is very shiny.
Figure 8: I recommend checking the size of white spaces between the photography. In D to F I suggest replacing the asterisk for head arrows, the septum will be more easy to identify.
Figure 9: I highly recommend edit this figure. The figure has many details that can be easily resolved. For example, a black bar under C label, the blue backgrounds between the photography is not equal, pinholes (F, G and I), and pinhead (H, F and I), different size of white spaces between the photography. The use of Photoshop or other software could be helpful to improve the figure.
*In Table 2:
The origin and insertion of the tensor fascia latae (TFL) are not mentioned in the table. Neither the muscle of the foot: plantaris profundus (PP), flexor digitorum brevis superficialis (FDBS), transversus plantae proximalis and distalis (TPP and D), and intertarsalis (IN), extensor digitorum communis longus (EDCL), abductor brevis dorsalis (ABD), tarsometatarsal (TMT).
Please check the name of TiAL1 and TiAL2. Also, you wrote “tibialis anterior brevis” instead of “tibialis anticus brevis”.

---

## Round 0.2 · Minor Revisions

I think that we are almost there. Please take these last comments of our reviewer into account so we can move forward. Thank you!

Reviewer 2 ·

Basic reporting

No comment

Experimental design

No comment

Validity of the findings

No comment

Additional comments

In revision, you adequately addressed my suggestions. I believe the paper is of high quality and deserves publication. However, I just have a few more minor comments.

Intro
Line 93. Delete “(e. g. Through Morphosource)” at the end of the paragraph or include statement in previous parenthesis.

Results

Just use abbreviations in the text as you have already referred the reader to Table 2 where muscles acronyms are listed.

Line 175. Change comma for colon after ‘pelvic muscles group’ as you are listing muscles.

Discussion

Line 394-395. ‘May have’ is repeated twice, delete one.

Table 1. I recommend deleting ‘test round’ in the headers and changing ‘Total stain duration’ for something like ‘Cumulative stain duration’.

Figure 1-9. Why is the scale bar quoted as being ‘approximately 1 cm’? You should get precise measurements from CT images.

---

## Round 0.3 · accepted · Accept

Your manuscript is ready to be published, congratulations, nice work!

#